# High-Level rAAV Vector Production by rAdV-Mediated Amplification of Small Amounts of Input Vector

**DOI:** 10.3390/v15010064

**Published:** 2022-12-24

**Authors:** Stefan Weger

**Affiliations:** Charité–Universitätsmedizin Berlin, Corporate Member of Freie Universität Berlin and Humboldt-Universität zu Berlin, Clinic for Neurology with Experimental Neurology, Gene Therapy Group, Campus Benjamin Franklin, Hindenburgdamm27, 12203 Berlin, Germany; stefan.weger@charite.de

**Keywords:** rAAV packaging, adeno-associated virus, recombinant adenovirus, RIS-Ad

## Abstract

The successful application of recombinant adeno-associated virus (rAAV) vectors for long-term transgene expression in clinical studies requires scalable production methods with genetically stable components. Due to their simple production scheme and the high viral titers achievable, first generation recombinant adenoviruses (rAdV) have long been taken into consideration as suitable tools for simultaneously providing both the helper functions and the AAV *rep* and *cap* genes for rAAV packaging. So far, however, such rAdV-*rep/cap* vectors have been difficult to generate and often turned out to be genetically unstable. Through ablation of *cis* and *trans* inhibitory function in the AAV-2 genome we have succeeded in establishing separate and stable rAdVs for high-level AAV serotype 2 Rep and Cap expression. These allowed rAAV-2 production at high burst sizes by simple coinfection protocols after providing the AAV-ITR flanked transgene vector genome either as rAAV-2 particles at low input concentrations or in form of an additional rAdV. With characteristics such as the ease of producing the required components, the straightforward adaption to other transgenes and the possible extension to further serotypes or capsid variants, especially the rAdV-mediated rAAV amplification system presents a very promising candidate for up-scaling to clinical grade vector preparations.

## 1. Introduction

Recombinant viral vectors based on different subtypes of the helper-dependent adeno-associated virus (rAAV) exhibit robust and long-term transgene expression in a wide range of target tissues. These properties have boosted the use of rAAV in gene transfer protocols for the treatment of a variety of genetic diseases such as hemophilia [1,2], metabolic disorders [3], congenital blindness [4,5] and muscular dystrophies [6], also highlighted by the recent clinical approvals of the AAV serotype 2 based vector Luxturna for the treatment of Leber’s congenital amaurosis [7] in 2017 and the AAV-9 vector Zolgensma for the treatment of spinal muscular atrophy [8] in 2019.

The use of rAAV vectors in the clinical setting, especially in the treatment of human diseases involving body-wide gene transfer, requires production and purification systems capable of generating large amounts of highly purified and infectious vector particles [9]. The original approach for rAAV production, the cotransfection of HEK293 cells with a mixture of plasmids providing the transgene cassette flanked by the AAV inverted terminal repeats (ITR), the AAV *rep* and *cap* genes and the essential helper virus functions [10,11,12] is still widely used for clinical grade vector preparations [9]. Whereas substantial progress has been made in recent years in adapting such transfection-based production systems to suspension cells [13], the transfection step may still be a limiting factor regarding vector yield and ease of handling. To establish exclusively infection-based rAAV production protocols, a variety of systems have been developed that combine the ITR-flanked transgene cassette, the AAV *rep* and *cap* genes and the required viral helper functions on one or several recombinant helperviruses. Such systems have been successfully implemented with herpes simplex virus (HSV-1) in mammalian [14,15,16] and baculovirus in insect cells [17,18,19]. For adenovirus (AdV), the most thoroughly characterized helper virus for productive AAV infection [20] however, recombinant viruses (rAdV) carrying the AAV *rep* and *cap* genes for a long time either could not be propagated at all or were genetically unstable [21]. In contrast, rAdV/AAV hybrid vectors carrying the AAV-ITR-flanked transgene have been successfully used in combination with Rep/Cap expressing cell lines for large-scale rAAV production [22].

Inhibition of adenoviral replication was also observed with rAdV hybrid vectors exclusively containing the AAV *rep* gene [23,24,25,26] and the failure to stably propagate rAdV/*rep-cap* vectors was therefore attributed to Rep protein expression. However, recent studies [27,28,29] have shown that the limited replication of these vectors is in large parts due to a *cis* inhibitory sequence in the 3′-part of the *rep* gene termed RIS-Ad (*Rep inhibition sequence for adenoviral replication*). It inhibits adenoviral genome replication by a transcription-associated mechanism, apparently mediated by promoter-proximal paused RNA II polymerase complexes [28].

Based on our earlier studies regarding the exact nature of the RIS-Ad, we generated stable rAdVs for separate high-level expression of the AAV-2 Rep and Cap proteins under the control of heterologous promoters. These rAdVs were then used for the production of rAAV vectors, with different strategies to provide the ITR-flanked transgene vector genome for packaging. Especially the rAdV-mediated amplification of small amounts of input rAAV vector seems most promising for up-scaling to preclinical and clinical requirements. With an optimized adenoviral vector for expression of the AAV-2 Rep proteins it reproducibly resulted in clearly higher yields of rAAV than the standard co-transfection procedure.

## 2. Materials and Methods

### 2.1. Plasmids and Cloning

Plasmid pTR-UF5 [30] carrying a CMV enhancer/promoter driven GFP cassette flanked by the AAV-2 ITR’s was kindly provided by Nick Muzyczka (University of Florida). The plasmids pShuttle-p5-Rep and pShuttle-p5-sRep containing the AAV-2 p5 promoter and the complete Rep78 coding region have been described [29]. The term *sRep* denotes a Rep78 sequence with recoding of AAV-2 nt 1782 to 1916 according to [27]. For generation of pShuttle-MMTV-Rep and pShuttle-MMTV-sRep, the sequences between the BglII site preceding the p5 promoter and the unique SalI site within the Rep-ORF (AAV-2 nt 1428) in the corresponding pShuttle-p5-Rep constructs were replaced by a BamHI-partial/SalI restriction fragment from pDG [11] containing the MMTV-LTR and the amino-terminal part of the Rep ORF. For cloning of pShuttle-M2-Tet-Rep constructs, a cassette containing the CMV promoter, the rtTA (M2) transactivator and the Tet promoter (an array of 7 Tet-Operator sequences followed by a CMV core promoter) was excised as HindIII/AgeI fragment from plasmid pD-Tet-M2-K1, and assembled with the corresponding Rep78 ORFs (either wildtype or sRep nucleotide sequence), generated as PCR fragments with flanking AgeI/XbaI sites, in HindIII/XbaI digested pShuttle vector by three-component ligation. For pShuttle-HSV-TK-sRep78, the M2-Tet cassette in pShuttle-M2-Tet-Rep was replaced by the HSV-TK promoter amplified from a suitable plasmid with primers generating HindIII and AgeI sites for cloning. In pShuttle-HSV-TK-sRep68 and pShuttle-HSV-TK-sRep-Stop531, the complete Rep78 ORF was replaced by the region up to the AAV-2 splice donor site with primers generating AgeI and XbaI (Rep-Stop531) or BssHII (Rep68) sites, respectively. The additional region in pShuttle-HSV-TK-sRep68 encoding the 7 Rep68 specific carboxy-terminal amino acids was introduced by a BssHII/XbaI linker. pShuttle-p40-Cap and pShuttle-s-p40-Cap have been described [29]. For expression of the AAV-2 Cap proteins under the control of heterologous promoters CMV (*human cytomegalovirus early enhancer/promoter*), PGK (*murine phosphoglycerate kinase promoter*), HSV-TK (*herpes simplex virus thymidine kinase promoter)* and adenovirus subtype 5 MLP (*major late promoter*), the p40 promoter sequences in pShuttle-p40-Cap up to the HindIII site at AAV-2 nt 1882 were replaced by the corresponding heterologous promoter sequences amplified from suitable parental plasmids with primers generating XbaI and HindIII sites for cloning. The Ad-MLP sequence contains nt -506 to +192 with respect to the start site for the late Ad5 transcripts including the first leader sequence, while the sequence denoted as Ad-MLP-I additionally contains the complete first intron and the second leader sequence (nt -506 to 1134 with respect to the transcription start site). For generation of pShuttle-ITR-GFP containing an CBA (chicken ß-actin CMV hybrid promoter) driven GFP gene flanked by the AAV-2 ITR’s, the complete ITR-flanked transgene cassette was excised from pTR-UF26 [19] with PstI, first subcloned into pBluescript II SK (+) and then excised with EcoRV/XbaI for cloning into the pShuttle vector. An additional PacI site present in a non-essential region of the transgene cassette was deleted by partial digestion, fill-in reaction and religation. All pAdEasy constructs were generated from the corresponding pShuttle plasmids through homologous recombination of PmeI linearized DNA in *E. coli* BJ5183-AD-1 cells (Agilent) containing the pAdEasy-1 plasmid (Genbank accession no. AY370909).

### 2.2. Cell Culture and Transfection

HEK293 (human embryonal kidney, ATCC CRL-1573) and HeLa (human cervix carcionoma) cells were propagated in Dulbecco’s modified Eagle medium (DMEM) supplemented with 10% fetal calf serum (FCS) and 100 µg/mL of penicillin and streptomycin at 37 °C with 5% CO_2_. Transfections of HEK-293 cells were performed by the calcium phosphate precipitate technique as described [31]. The numbers of cells seeded the day before transfection were 3 × 10^5^ cells for 6-well plates, 6 × 10^5^ cells for 25 cm^2^ flasks or 4 × 10^6^ cells for 14.5 cm dishes, respectively.

### 2.3. Generation and Propagation of Recombinant Adenoviruses (rAdV)

Generation of rAdV was performed with the AdEasy system [32,33] largely as described. The corresponding pShuttle constructs were linearized with PmeI and electroporated into *E.coli* BJ5183-AD-1 cells for homologous recombination with the E1/E3 deleted Ad5 genome contained on the stably transformed pAdEasy plasmid. Recombinant pAdEasy plasmid DNA was amplified in *E.coli* XL-1-Blue cells, linearized with PacI and transfected into HEK-293 cells in 25 cm^2^ flasks. The primary rAdV preparation was harvested 12 to 14 days post transfection by four freeze–thaw cycles in liquid nitrogen/37 °C water bath in 2 mL of medium. 1.5 mL of the supernatant was used for first round of amplification with 2 × 10^6^ cells seeded in 75 cm^2^ flasks the day before infection. Further rounds of amplification and purification by CsCl gradient centrifugation were performed as described [33]. Essentially, 1 × 10^7^ cells in 175 cm^2^ flasks were infected with 2.5 × 10^10^ genomic rAdV particles from the previous round for 44 to 50 h and lysed in 5 mL of residual medium by three freeze–thaw cycles.

### 2.4. Quantification of Genomic rAdV Particles

Adenoviral genomic particles in rAdV freeze–thaw supernatants or CsCl purified rAdVs were determined by Light-Cycler based Real Time PCR with the GoTaq qPCR Master Mix (Promega) using Primers E4-Q1 and E4-Q2 for amplification of a 202 bp region from the Ad5-E4 region as described recently [29].

### 2.5. Infection of HEK-293 Cells with Recombinant Adenoviruses (rAdV)

For western or Southern blot analysis and production of rAAV, HEK-293 cells were infected in the appropriate cell culture vessels with the indicated amounts (MOI generally stated in genomic particles per cell) of rAdV freeze–thaw lysates or CsCl purified rAdVs without prior removal of the cell culture media.

### 2.6. Western Analysis

Immunoblot analysis of purified rAAV-2 preparations for capsid content or of whole cell extracts from transfected or rAdV infected HEK-293 cells for AAV-2 Cap or Rep expression was performed with anti-Cap monoclonal antibody B1 diluted 1:20 or anti-Rep monoclonal antibody 303.9 diluted 1:10 (both from Progen, Heidelberg, Germany), respectively, followed by incubation with horseradish peroxidase conjugated secondary antibodies and enhanced chemiluminescent detection (ECL) as described previously [31].

### 2.7. Silver Staining

Capsid protein contents of purified rAAV-2 vector preparations were analyzed in silver stained gels with different amounts of vector, corresponding to 1.0 × 10^8^ to 1.0 × 10^10^ genomic particles, directly lysed in protein sample buffer, incubated for 5 min at 95 °C and separated on 8% polyacrylamide/SDS gels. The staining reaction was performed as described in [19].

### 2.8. Extraction of Viral DNA and Southern Hybridization

Extraction of viral DNA by a modified Hirt procedure was performed essentially as described [31]. Southern Blot analysis was performed with 6 µg of Hirt-DNA’s digested with 40 units of *DpnI* for 2 h. After electrophoresis on a 0.8 % agarose gel, DNA’s were transferred by capillary blotting to a nylon membrane (Hybond-N, GE Healthcare) and hybridized with a Biotin-11-dUTP-labeled 1.9 kb EcoRI probe from pTR-UF5 comprising the CMV promoter, the GFP-ORF and the SV40 polyadenylation site. The bound biotinylated probe was detected with Streptavidin horseradish peroxidase conjugate (high sensitivity HRP Conjugate, Thermo Scientific) followed by enhanced chemiluminescent detection (ECL).

### 2.9. Production and Purification of rAAV in HEK-293 Cells

HEK-293 cells were seeded at 25 to 30% confluency. 24 h later, cells were either (I) co-transfectd with the pDG helper and the pTR-UF5 GFP-expressing vector plasmid at a 1:1 molecular ratio as positive control, (II) transfected with pTR-UF5 and additionally infected with the two rAdVs for Rep and Cap protein expression, (III) transduced with prepackaged rAAV-GFP and co-infected with the two rAdVs for Rep and Cap protein expression or (IV) triple-infected with rAdV-ITR-GFP providing the ITR-flanked GFP vector DNA and the two rAdVs for Rep and Cap protein expression. For the approaches involving transfection the cell medium was replaced by fresh medium 16 h post transfection. Cells were generally harvested 64 h post transfection/infection. For optimization of rAAV packaging in 6-well plates, the scraped cells were directly lysed in the medium by three freeze–thaw cycles and the supernatant after centrifugation stored at −80 °C fur further use. For medium scale rAAV production and purification, harvested cells were washed with PBS, resuspended in 1 mL of lysis buffer (10 mM Tis-HCl pH 8.5, 150 mM NaCl, 1 mM MgCl_2_ and 1% Triton-X-100 v/v) per 14.5 cm dish and lysed by three freeze–thaw cycles. Crude lysates were treated with 250U benzonase (Merck) per ml lysate at 37 °C for 1 h to degrade unpackaged AAV DNA and centrifuged at 8000× *g* for 30 min to pellet cell debris. The supernatant was frozen at −80 °C for at least 1 h. After thawing and centrifugation at 8000× *g* for 15 min the supernatant was subjected to one-step AVB sepharose affinity chromatography using 1 mL prepacked HiTrap columns on an ÄKTA purifier system (GE Healthcare) as described [19].

### 2.10. Quantification of rAAV Vector Preparations

Aliquots of affinity purified rAAV-GFP vector preparations were digested with proteinase K (Roth, Germany) at a final concentration of 400 µg/mL at 56 °C for 2 h in lysis buffer (1% [w/v] N-Lauroylsarcosine, 25 mM Tris pH 8.5, 10 mM EDTA pH 8.0) to release the vector genomes from the capsids. Following phenol-chloroform extraction and ethanol-precipitation, samples were dissolved in 20 µL of Tris-EDTA buffer pH 7.6 and prediluted 1:100 for the determination of genomic copy numbers by Light-Cycler based Real Time PCR with the GoTaq qPCR Master Mix (Promega). Primers specific for the bovine growth hormone derived polyA site of the vector backbones were used as described [19]. To assay possible wildtype AAV-2 contaminations of the purified rAAV vector preparations, a Light-Cycler based Real Time PCR with primers amplifying the AAV-2 p5 promoter region (AAV-2 nucleotides 191 to 310) was performed. This region was not contained in any of the rAdVs used for expression of the AAV-2 Rep or Cap proteins. The detection limit of the PCR assay was in the range of 10 genomic AAV-2 copies per 5 µL sample corresponding to 2 × 10^3^ genomic particles/mL.

### 2.11. Determination of rAAV-GFP Transduction Efficiencies in HeLa Cells by FACS Analysis

2 × 10^5^ HeLa cells in 12-well plates were either transduced with small volumes of freeze–thaw supernatants from HEK-293 cells used for rAAV-GFP packaging (usually a volume corresponding to 1/15 of the original HEK-293 6-well was applied) or with affinity purified rAAV-GFPs at the indicated MOIs (vg/cell), and superinfected with adenovirus type 2 at a MOI of 10 (infectious titer). Cells were harvested 44 h post transduction, washed once with PBS and resuspended in 500 µL of 20% FCS in PBS. FACS analysis was performed with a FACS Calibur according to the manufacturer’s protocol (Becton & Dickinson). The cut-off level for GFP-positive cells was determined with non-transduced, Ad2 infected HeLa cells as controls.

## 3. Results

### 3.1. Generation of Recombinant First Generation Adenoviruses (rAdV) Expressing AAV-2 Rep and Cap Proteins

#### 3.1.1. General Strategy

Based on the results of previous studies [27,28], recoding of the AAV-2 RIS-Ad appeared as a promising strategy for the expression of functional AAV-2 Rep proteins through first generation recombinant adenoviral vectors (rAdV). The RIS-Ad is a regulatory function located in the 3′-part of the AAV-2 rep gene (Figure 1A), which strongly inhibits adenoviral replication *in cis*. In the context of the complete AAV-2 genome however, recoding of the RIS-Ad would restrict expression of the AAV-2 cap proteins through loss of p40 promoter activity and splicing at the major splice donor site [28]. For rAdV-mediated expression of both the non-structural and structural of AAV-2, the Rep and Cap ORFs were therefore separated onto two individual rAdVs.

#### 3.1.2. Generation of rAdVs for Rep78/Rep52 Expression

The recoded region in the rAdV-Rep vectors (termed *sRep*) comprised Rep78 amino acids 488 to 531, corresponding to AAV-2 nucleotides 1782 to 1916 (Figure 1A,B). As recoding impairs splicing, these vectors were expected to express mainly Rep78 and Rep52 proteins, but no Rep68 or Rep40 proteins translated from spliced transcripts. Due to the redundancy in Rep protein function, however, expression of one large and one small Rep protein should be sufficient to promote rAAV packaging. In addition to the *cis* inhibitory effect of RIS-Ad, the Rep78 protein has also been shown to inhibit adenoviral replication *in trans* [29,34]. Besides the physiological AAV-2 p5 promoter, the rather weak MMTV-LTR promoter and a tetracycline inducible expression system (Tet-On) were therefore tested for their suitability to drive Rep78 expression in the adenoviral vectors (Figure 1B, p5, MMTV and M2-Tet Rep constructs). Rep52 was generally expressed from the internal p19 promoter (Figure 1B). In line with previous studies, all adenoviral shuttle constructs harboring the non-recoded RIS-Ad Rep sequence did not give rise to rAdVs in HEK293 cells in repeated experiments (Figure 1B). The recoded adenoviral p5-sRep construct led to low titer rAdV preparations expressing small amounts of the Rep proteins in the initial amplification rounds, but loss of viral titers and Rep protein expression were observed in later rounds (indicated by “(+)“ in Figure 1B). In contrast, the rAdV preparations obtained after transfection of recoded MMTV-sRep and M2-Tet-sRep adenoviral shuttle constructs could be successfully passaged with titers similar to those of a rAdV-CMV-GFP control vector. After infection of HEK293 cells with round 6 supernatants of the two rAdVs at different MOIs (as indicated in Figure 1C), Rep expression levels were compared to those obtained after transfection of pDG (transfection efficiencies in the range of 70 to 80%). pDG is a well-established standard plasmid for rAAV-2 vector production [11], which expresses the AAV-2 Rep and Cap proteins and the adenoviral helper function E2a, E4orf6 and VA-RNA. It was generally used as positive control in the present study. Already at vector doses of 400 to 1000 vg/cell, both rAdVs showed similar Rep52 levels as the positive control (Figure 1C). Compared to pDG transfection, however, rAdV-MMTV-sRep showed lower Rep78 expression levels (except for the highest MOI of 4000 vg/cell), whereas rAdV-M2-Tet-sRep showed similar or slightly higher Rep78 levels, depending on the exact MOI (Figure 1C). The large Rep78 protein expressed from rAdV-M2-Tet-sRep was functional in promoting replication of an AAV-2 ITR flanked GFP vector cassette provided by transfection (plasmid pTR-UF5, Figure 1E). However, at all MOIs tested, abundances of replicative intermediates (Figure 1E, monomeric RF_M_ and dimeric RF_M_) were lower than after cotransfection of pDG with pTR-UF5 (Figure 1E, lane 2 marked “pos“). Rather unexpectedly, neither expression of Rep78 (Figure 1D) nor replication of the rAAV vector genome (Figure 1E) was increased in the presence of doxycycline (a non-hydrolysable tetracycline analogue) with rAdV-M2-Tet-sRep infection. Therefore, all subsequent experiments were performed in the absence of inducer. With the rAdV-MMTV-sRep vector expressing the large Rep78 protein under control of the MMTV LTR, levels of replicative intermediates were even further reduced as compared to rAdV-M2-Tet-sRep.

#### 3.1.3. rAdVs Expressing Carboxy-Terminal Truncated Rep Proteins under HSV-TK Promoter Control

As shown in the previous subsection, expression of the large Rep78 protein was obviously a limiting factor for both, rAdV-M2-Tet-sRep and rAdV-MMTV-sRep, in promoting replication of an AAV-2 ITR flanked vector genome. Based on promising results for rAdV-mediated expression of the AAV-2 Cap proteins (see subsequent Section 3.1.4), the herpes simplex virus thymidine kinase (HSV-TK) promoter was therefore tested as means to increase the ratio of large to small Rep proteins expressed from the rAdV-Rep vectors with the corresponding small Rep protein remaining under control of the physiological p19 promoter (Figure 2A). To additionally abolish the trans inhibitory effects on adenoviral replication described for the AAV2 intron region [35], this region was completely deleted (Figure 2A). Two constructs differing only in the presence or absence of the seven Rep68/Rep40 specific carboxy-terminal amino acids were generated, with both recoded downstream from Rep78/Rep68 amino acid 488 (Figure 2A, sRep68 and sRepStop531). The corresponding rAdVs led to a higher proportion of the corresponding large Rep proteins (Rep68 or RepStop531, respectively) as compared to the rAdV-M2-Tet-sRep vector (Figure 2B). In line with this finding, they also promoted a clearly stronger DNA replication of a recombinant rAAV genome, which in this experiment was provided by co-infection with a purified rAAV2-GFP vector (Figure 2C).

#### 3.1.4. Generation of rAdV for High-Level AAV-2 Cap Expression

The wildtype p40 promoter constitutes a major element of the RIS-Ad (compare Figure 1A) and as already demonstrated in earlier studies [29], no rAdVs expressing the AAV-2 Cap proteins could be generated using this promoter (Figure 3A). Therefore, a variety of heterologous promoters including the CMV immediate early promoter, the cellular phosphoglycerate kinase (PGK) promoter and the HSV-TK promoter as well as two version of the adenovirus subtype 5 major late major late promoter (MLP, differing in the presence or absence of the first intron) were tested as alternatives for rAdV-mediated Cap expression (Figure 3A). In the corresponding adenoviral shuttle plasmids, the respective heterologous promoter was fused to the cap gene at the HindIII site 18 bp upstream of the splice donor site. Only those constructs containing the HSV-TK or a mutated AAV-2 p40 promoter comprising the recoded sRep (s-p40) region gave rise to rAdVs. As expected, rAdV-s-p40-Cap displayed hardly detectable VP expression levels and therefore only rAdV-HSV-TK-Cap was further pursued as means for providing the capsid proteins for rAAV packaging. Western blot analysis of protein extracts from HEK293 cells infected with rAdV-HSV-TK-Cap preparations from amplification rounds 3 to 8 showed high-level expression of the three capsid proteins VP1 to VP3 (Figure 3B) with a stoichiometry comparable to that of the positive control (pDG transfection, lane 2 in Figure 3B). Saturation of capsid proteins levels was observed at rAdV vector doses of about 1000 vg/cell (Figure 3C).

### 3.2. Optimization of rAdV-Mediated rAAV Vector Packaging

Three different possible setups for rAdV-mediated rAAV vector packaging were then first optimized in small-scale experiments in HEK293 cells (Figure 4). They all involved co-infection with the newly generated rAdVs for AAV-2 Rep and Cap expression, but differed in the mode of providing an AAV-2 ITR flanked vector genome harboring a CMV-GFP transgene cassette: (I) pTR-UF5 transfection, (II) rAdV triple infection with an additional rAdV containing the ITR-flanked GFP transgene or (III) amplification of prepackaged rAAV-GFP vectors provided at low MOIs.

Packaging efficiency was monitored by transduction of HeLa cells with small aliquots of freeze–thaw lysates of the packaging reactions and determination of GFP positive cells (Figure 5). Compared to rAdV-M2-Tet-sRep and rAdV-HSV-TK-sRepStop531, infection with rAdV-MMTV-sRep led to only low numbers of transducing rAAV-GFP vector particles and was therefore excluded from further analysis. Representative experiments for each of the three setups described above are shown in Figure 5A–C for the combination of rAdV-M2-Tet-sRep and rAdV-HSV-TK-Cap. As positive control for rAAV-2 GFP packaging, the standard protocol of cotransfecting the vector plasmid pTR-UF5 with the helper/packaging plasmid pDG was used. With optimized rAdV MOIs, very similar amounts of transducing particles as in the positive control were obtained for all three setups (Figure 5A–C). Similar to the pTR-UF5 transfection/rAdV infection protocol (Figure 5A), the rAdV triple infection (compare Figure 4) led to high yields of transducing rAAV particles with a clear decrease at higher MOIs of the rAdV-ITR-GFP vector (Figure 5B). This is probably due to interference between the three rAdVs leading to lower Rep and Cap expression as demonstrated by Western analysis. In addition we encountered a drop in rAAV packaging efficiencies with increasing amplification rounds. As exclusively infection based rAdV-mediated rAAV packaging systems of choice, we therefore favored the rAAV amplification protocol (protocol III in Figure 4). The amount of rAAV-GFP transducing units could be amplified by a factor of up to 200 in a single step through co-infection with rAdV-M2-Tet-sRep and rAd-HSV-TK-Cap, although quite high MOIs of rAdV-M2-Tet-sRep (4000 to 8000 genomic particles per cell) were required for optimal yields (Figure 5C). The rAAV amplification protocol was also optimized as first choice for the rAdVs expressing the carboxy-terminal truncated sRep68 and sRep-Stop531 proteins under control of the HSV-TK promoter (compare Section 3.1.4). Over a wide range of MOIs, high yields of transducing rAAV-GFP vector could be obtained with rAd-HSV-TK-sRepStop531, with an optimum already at 200 to 400 genomic particles per cell (Figure 5D, smaller aliquots of packaging reactions were used for transduction than in Figure 5C). Similar results were obtained with rAd-HSV-TK-sRep68, but this rAdV did not prove as stable as rAd-HSV-TK-sRepStop531 (see Section 3.4).

### 3.3. Yields and Characterization of rAAV Vector Preparations Generated by rAdV Co-Infection

To more closely analyze rAAV vectors generated with the novel rAdV-mediated packaging systems, several independent (individual numbers are indicated in the figure legend to Figure 6A) medium scale rAAV-GFP vector preparations were performed under the optimized conditions for each of the setups described in Section 3.2 (compare Figure 4 and Figure 5).

rAAV-GFP vectors were harvested by freeze–thaw lysis from 6.6 × 10^7^ HEK293 cells initially put into the production process and subjected to one-step HPLC-based AVB affinity purification. In agreement with the data from the optimization experiments, the yields of purified rAAV-GFP vector fractions for the positive control (pTR-UF5/pDG co-transfection), the pTR-UF5 transfection followed by rAdV-M2-Tet-sRep/rAdV-HSV-TK-Cap co-infection, the rAdV triple infection and the rAAV-GFP amplification mediated by co-infection with the rAdV-M2-Tet-sRep/rAdV-HSV-TK-Cap combination (Figure 6A, first four bars) were all in a similar range of about 5.0 to 7.5 × 10^3^ vector genomes/input cell (vg/cell, the so-called burst size). Replacing rAdV-M2-Tet-sRep in the rAAV-GFP amplification protocol with rAdV-HSV-TK-sRepStop531 led to clearly elevated burst sizes in the range of 1.5 to 2.8 × 10^4^ vg/cell (Figure 6A, most right bar). Thus the rAAV vector amplification with the combination of rAdV-HSV-TK-sRepStop531 and rAdV-HSV-TK-Cap was clearly superior to the other setups including the co-transfection protocol and led to an up to 1400-fold amplification of the input amount of rAAV-GFP genomic vector particles. AAV-2 wildtype genomes were below the detection limit of the PCR-based assay in all preparations. Whereas the vectors produced by rAdV-mediated rAAV amplification showed similar transduction rates as the positive control in HeLa cells over a wide range of MOIs, those produced by pTR-UF5 transfection/rAdV infection or by rAdV triple infection showed a trend towards higher transduction efficiencies at lower MOIs of genomic particles (Figure 6B). Capsid content, composition and integrity of the different rAAV-GFP preparations were judged by SDS page of defined numbers of purified genomic particles followed by silver staining. Background-free VP1, VP2 and VP3 bands with apparently similar ratios of the individual capsid proteins were detected for all preparations (Figure 6C,E) with the identities of the three major silver-stained bands confirmed in anti-Cap Western blot analysis (Figure 6D,F). All rAAV vector preparations involving infection with rAdV-M2-Tet-sRep showed a higher capsid protein content than the one generated by the classical co-transfection protocol, especially evident for the pTR-UF5 transfection/infection and the rAdV triple infection protocol (Figure 6C,D). Based on densitometric scanning of the silver-stained gels, the rAAV-2 preparations involving rAdV-M2-Tet-sRep mediated amplification of input rAAV-2 vector had a 4-fold higher capsid content than those from the positive control when normalized to the number of genomic particles. Those from the pTR-UF5 transfection/rAdV infection and those from the rAdV triple infection even had a 13-fold and about 50-fold higher capsid content, respectively. Thus these preparations contain a large excess of empty to full capsids as compared to the standard co-transfection protocol. The ratio of full to empty capsids was greatly improved when using rAdV-HSV-TK-sRepStop531 for rAAV packaging instead of rAdV-M2-Tet-sRep. Two representative rAAV-GFP vector amplifications had an only 2-fold higher (Figure 6E, left side) or even 2.5-fold lower (Figure 6E, right side) capsid content as the respective positive control preparations with the identity of the individual capsid bands again confirmed by Western blot analysis (Figure 6F).

### 3.4. Novel rAdVs for AAV-2 Rep and Cap Expression Are Highly Stable

An important requirement for using rAdVs in reproducible packaging of rAAV vectors is genomic stability. The two adenoviral vectors rAdV-M2-Tet-sRep and rAdV-HSV-TK-Cap initially used were therefore passaged over a total of 15 rounds. Genomic titers obtained in freeze–thaw lysates were in a constant range of 2 to 6 × 10^11^ vg/mL for both rAdVs after few rounds of amplification (Figure 7A) and total numbers of rAdV genomic particles could be raised by a factor of 50 to 150 in each individual amplification step.

Assessment of the integrity of the vector genome by PCR-based analysis with primers spanning the complete transgene cassettes revealed only one distinct respective major PCR-amplification product throughout all rounds of amplification (Figure 7B). These bands corresponded in size to the respective full-length sequences, as also demonstrated by comparison with the PCR products obtained from the corresponding adenoviral shuttle plasmids (positive controls in Figure 7B). No lower-sized bands pointing to possible deletions of parts of the transgene cassettes in subpopulations of the rAdV vector preparations were observed. Rep and Cap protein levels, respectively, also remained constant throughout the higher rAdV amplification rounds. Furthermore, when comparing the ability of the rAdV-M2-Tet-sRep preparations from amplification rounds 4 and 15 to support rAAV2-GFP vector packaging in small and medium scale setups, only minor differences within the range of variation of the experimental setup were found. Two rAdV vectors with intron-deleted large Rep proteins (Rep68 and RepStopStop531) under control of the HSV-TK promoter had been initially generated (as described in Section 3.1.3). In comparison to rAdV-M2-Tet-sRep78, they both promoted strongly enhanced replication of a recombinant AAV genome (compare Figure 2C). In initial small-scale experiments for optimization of the MOIs for mediating efficient rAAV packaging, a sharp drop in the capability of rAdV-HSV-TK-sRep68 to promote rAAV packaging was observed in preparations of this vector amplified beyond round 5. This was accompanied by a loss in Rep expression levels, but not in rAdV titers. PCR-analysis of the rAdV-HSV-TK-sRep68 transgene cassette with primers located in the flanking adenoviral sequences revealed an additional band about 1.0 kb larger than the expected size of 2.5 kb, which became prominent starting with round 6 (Figure 7C). This points to possible insertions within the transgene cassette. In contrast, for rAdV-HSV-TK-sRep-Stop531, only the band of the expected size was seen up to round 10, the highest amplification round investigated (Figure 7D). As described, all subsequent packaging experiments were therefore performed with rAdV-HSV-TK-sRep-Stop531. In summary, all rAdVs used for the medium-scale packaging and affinity purification of rAAV-GFP vectors described in Section 3.3 exhibited a high genomic stability and a stable Rep or Cap transgene expression.

## 4. Discussion

The successful application of AAV based vectors for the treatment of a variety of human genetic diseases has led to an increasing demand for rAAV large-scale production systems. Pre-clinical studies for toxicology, safety, dose, and bio-distribution assessments alone may often require up to 1E15 to 1E16 vector genomes, especially for systemic application in large animal models [9]. Whereas many vectors used in clinical trials are still produced by suspension cell adapted classical plasmid co-transfection protocols, exclusively infection-based rAAV packaging systems may be advantageous in terms of easy handling and the lack of requirement for large amounts of highly purified plasmid DNA [9,13]. Although adenovirus (AdV) presents the prototypical [20] and best-characterized AAV helper virus, rAAV packaging systems based exclusively on first generation recombinant adenoviruses (rAdV) have long struggled with limited replication and stability of the rAdVs expressing the AAV Rep proteins [21].

One of the main keys to stable rAdV-mediated Rep protein expression was the partial recoding of the RIS-Ad, a *cis* inhibitory sequence in the 3′-part of the *rep* gene [27,28]. In AAV-2 replication, this region possesses a dual function as part of the Rep coding region on one side and as control region for p40 promoter driven *cap* transcription on the other side. The Rep and Cap ORFs were therefore allocated to separate rAdVs. This approach additionally offers a high flexibility with regard to adjusting the relative Rep and Cap expression levels and will facilitate the adaption of the system to non AAV-2 serotype vectors or capsid variants. In addition to the RIS-Ad acting as a *cis* inhibitory sequence, especially the large Rep protein Rep78 may limit adenoviral replication *in trans* [29,34]. For the initial adenoviral constructs for Rep78 and Rep52 expression, promoters upstream of the Rep78 start codon expression were therefore chosen with regard to low (MMTV-LTR) or tightly regulated (M2-Tet) expression. With the classical co-transfection system, reduced Rep78/Rep68 expression under control of the MMTV-LTR has even been shown to be advantageous for the production of recombinant AAV particles [11]. The validity of these considerations was confirmed by our failure to stably propagate a rAdV expressing the recoded Rep78 under p5 promoter control, although this limited stability may also involve interactions of Rep78 with multifunctional Rep binding elements in the p5 region [36]. The ratio of Rep78 to Rep52 in rAdV-M2-Tet-sRep was similar to that seen for the pDG plasmid optimized for the rAAV-2 packaging cotransfection protocol [11] used as positive control throughout our studies. However, DNA replication of a recombinant AAV genome by rAdV-M2-Tet-sRep seemed to be reduced as compared to the positive control. Not quite in line with the theoretical expectations, Rep78 expression levels from rAdV-M2-Tet-sRep could not be further increased by the addition of doxycycline. This was in clear contrast to the strong induction observed after transient transfection of the original pShuttle constructs used for the homologous recombination step for production of adenoviral DNA in *E.coli*. Obviously the minimal CMV promoter, which constitutes part of the tetracycline response element (TRE), may be stimulated by early adenoviral functions already in the absence of doxycycline, leading to elevated basal expression levels. The rAdV-MMTV-sRep vector showed only low Rep78 expression levels and was not as effective in promoting rAAV-DNA replication and packaging as rAdV-M2-Tet-sRep. It was therefore not further pursued in the later parts of the study.

Substantial further improvements of rAdV-mediated rAAV packaging were achieved by using the HSV-TK promoter to express large Rep proteins, which were devoid of sequences encoded by the AAV-2 intron. These Rep68 or Rep-Stop531 proteins, with the later terminating directly upstream of the AAV-2 intron, were fully functional in supporting rAAV-DNA replication and could be expressed at high levels in the context of rAdVs. AAV-2 intron encoded protein domains have been shown to suppress adenoviral replication by inhibiting PKA activity through direct protein–protein interactions [35]. In line with these findings, we were unable to stably propagate a rAdV with a recoded Rep78 protein expressed under control of the HSV-TK promoter. For the rAdV-HSV-TK-sRep68 vector containing 7 additional carboxy-terminal amino acids as compared to rAd-HSV-TK-sRep-Stop531, we observed a loss of Rep expression and support of rAAV packaging starting with amplification round 6. Quite surprisingly, no deletion of parts of the transgene cassette occurred during the amplification, but a larger band appeared in the PCR-analysis, pointing to possible duplications of transgene sequences. The phosphorylation of Rep68 at one of the unique 7 carboxy-terminal amino acids mediates its interaction with members of the 14-3-3 protein family and a Rep68-S535A point mutant is more efficient in promoting AAV DNA replication than the wildtype protein [37]. We have not elucidated yet, whether the putative duplication(s) within the rAd-transgene cassette may be due to such an interaction. However, we have firmly established that the partly recoded sRep-Stop531 protein missing the 7 Rep68 unique amino acids was fully functional for rAAV DNA replication and could be stably propagated within a rAdV.

To routinely use rAdVs in large-scale rAAV vectors packaging, these must possess a high genomic stability throughout multiple amplification steps. Of note, for a rAdV vector containing the combined wildtype Rep and Cap expression cassette, a partial loss of this cassette within only a few amplification rounds was reported [21]. In contrast, our rAdV vectors were completely stable for genomic integrity and transgene expression over at least 15 amplification rounds in case of rAdV-M2-Tet-sRep and rAdV-HSV-TK-Cap and for at least the 10 amplification rounds examined in case of rAdV-HSV-TK-sRep-Stop531.

Another key component of our novel system was the use of the HSV-TK promoter for rAdV-mediated expression of AAV-2 functions. Of a series of cellular and viral promoters tested, it was the only one found to lead to stable and high-level expression of the AAV-2 capsid proteins and it also proved suitable for expression of functional large Rep proteins. A prominent feature of the HSV-TK promoter is its almost exclusive regulation by two separate Sp1 transcription factor binding sites [38,39,40]. Of note, SP1 binding sites are also important for Rep-mediated induction of the AAV-2 p19 and p40 promoters in the presence of adenovirus [41,42]. Future studies will therefore address the question whether promoters mainly dependent on Sp1 sites generally may be well suited for rAdV-mediated expression of AAV functions.

One of the major drawbacks of our initial rAdV-M2-Tet-sRep based rAAV packaging systems appeared to be the large excess of empty over genome containing rAAV particles, as judged by capsid content. Empty particles have the potential to increase innate and adaptive immune responses after vector application [43]. Downstream purification of clinical grade vector preparations usually involves removal of such empty particles, e.g., by ion-exchange chromatography [44]. Importantly, the advanced system involving rAdV-HSV-TK-sRep-Stop531 led to a higher replication of the rAAV genome and resulted in similar or even lower proportions of empty particles than the standard co-transfection protocol. Of note, a certain amount of empty particles may even be beneficial for rAAV transduction in vivo. An up to 100-fold excess of empty capsids serving as decoy was used to overcome preexisting B-cell mediated humoral immunity, a major hurdle to systemic rAAV delivery, without loss of transduction efficiency in factor IX gene therapy [45]. In general, the expression of Rep and Cap from two separate rAdVs offers the possibility to adjust the capsid levels and thereby most probably also the proportion of empty particles over a wide range.

With the initial rAd-M2-Tet-sRep vector, burst sizes for the different rAd-based rAAV production schemes after affinity purification, taking into consideration the peak fractions of the purification only, were in a range of 5.0 × 10^3^ to 1.0 × 10^4^ vg/cell and thus very similar to the standard plasmid co-transfection system at high transfection efficiencies. The improved system using rAdV-HSV-TK-sRepStop531 for amplification of a pre-purified rAAV vector resulted in about three times higher yields of 1.5 × 10^4^ to 3.0 × 10^4^ vg/cell after affinity purification. Optimized rHSV-based rAAV production systems presently exhibit maximal burst sizes from 8 × 10^4^ to 1 × 10^5^ vg/cell in crude cell lysates or supernatants [16,46], resulting in final yields after purification of large-scale preparations of about 2.0 × 10^4^ vg/cell [9,46]. The original *Sf9* insect cell-based rAAV production system employing several recombinant baculoviruses (rBac) for Rep/Cap expression and delivery of the transgene [18] routinely results in rAAV burst sizes of 2 × 10^4^ vg/cell in bioreactor scale productions [17]. With *Sf9* cell lines containing the AAV *rep* and *cap* genes stably integrated into the host genome, rAAV-2 yields of up to 1.4 × 10^5^ vg/cell after iodixanol gradient centrifugation [47] and 9 × 10^4^ vg/cell in crude cell lysates [48] have been described. Despite not much obvious differences in vector yields in comparison to the well established rHSV- and Baculovirus based systems for large-scale rAAV production, we nevertheless reckon that our novel rAdV-based protocol may present an attractive alternative. One of the main advantages as compared to the rHSV system is the straightforward amplification of the rAdV helper viruses leading to much higher titers as compared to the ICP27 deleted rHSV vectors, which have to be complemented by a corresponding cell line [14]. The major challenges remaining with the insect cell based systems are the high particle-to-infectivity ratios reported. These may be due to non-optimal ratios of the three capsid proteins [18,49] or differential posttranslational modifications of the AAV capsid proteins as compared to mammalian cells [50].

Amplification of small amounts of input rAAV vector by rAdV co-infection offers several advantages as compared to providing the AAV ITR-flanked transgene cassette by additional helper viruses or stable cell lines. In addition to reducing manufacturing complexity, transgene cassettes with novel therapeutic genes and/or control elements can be put into a large-scale production process directly after cloning into an AAV-ITR-based vector plasmid and an initial characterization. Of note, a similar rAAV amplification system based on rAdVs for the combined expression of rAAV Rep and Cap functions was published very recently [51] and an additional benefit of rAAV amplification was found to be reduced reverse packaging of adenoviral sequences or antibiotic resistance genes.

A potential disadvantage of the novel rAdV-mediated amplification system to produce rAAV vectors for clinical applications may be contamination of the final rAAV preparation with infectious rAdV or adenoviral proteins, which can trigger strong immune responses in humans [52]. Several clinical rAAV lots have been produced using HeLa producer cell lines and Ad helper virus infection without any adverse events in the clinical application [9]. Nevertheless, with rAdV-based rAAV packaging systems, a special emphasis has to be put on the effective downstream purification procedures such as combinations of affinity and ion exchange chromatography to purify off any adenoviral contamination [53]. It may also be desirable to develop helper rAdV deficient for propagation in the cell line used for rAAV amplification as demonstrated by Su and co-workers [51].

## 5. Conclusions

As demonstrated by our study, the separate expression of high-levels of the AAV2 Rep and Cap proteins from two stable first-generation recombinant adenoviruses through elimination of cis and trans inhibitory elements in the AAV genome is feasible. The use of these rAdVs in amplification of small amounts of input rAAV vector holds great promise for large-scale rAAV vector production required for pre-clinical and clinical studies in terms of cost-effectiveness and easy handling. The required rAdVs can be easily propagated to high titers in standard HEK293. Ongoing work now focuses on the adaption of this packaging system to HEK293 suspension cells and extending it to further rAAV serotypes and capsid variants.

## Figures and Tables

**Figure 1 viruses-15-00064-f001:**
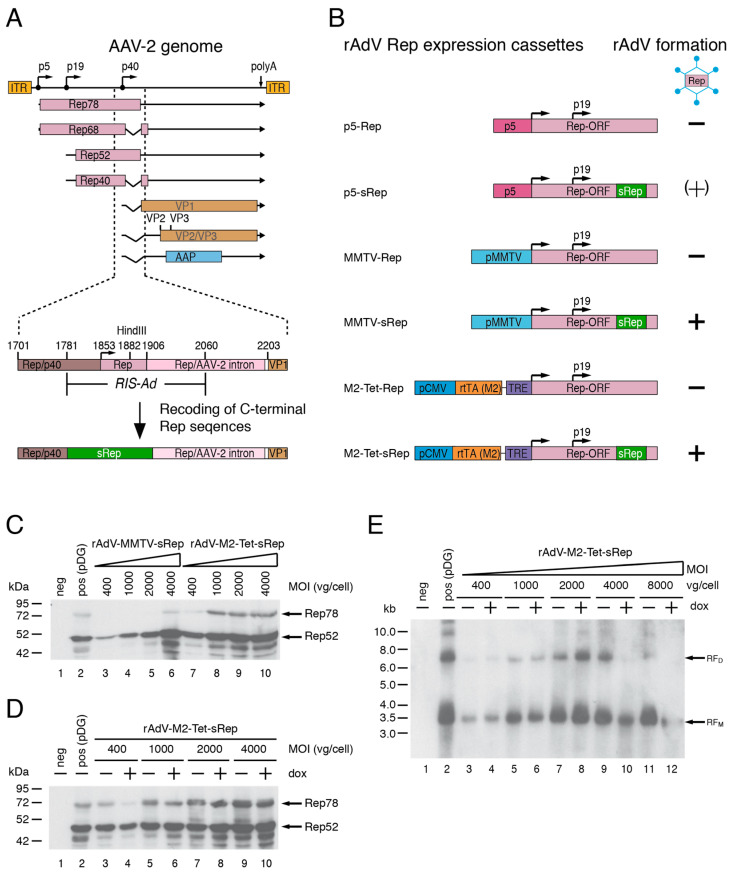
Expression of AAV-2 Rep proteins by first generation recombinant adenoviruses (rAdV) after recoding of the 3′-inhibitory RIS-Ad sequence. (**A**) Genome organization of AAV-2 with a close-up of the RIS-Ad (*Rep inhibitory sequence for adenoviral replication*) in the 3′-part of the rep gene. The AAV-2 inverted terminal repeats (ITRs), the four Rep proteins Rep78, Rep68, Rep52, Rep40, the capsid proteins VP1 to VP3 and the AAP (*assembly activating* protein) are indicated by differentially colored boxes. The box “sRep“ in the lower part of the figure indicates recoding of AAV-2 nucleotides 1782 to 1916 sufficient for abolishing the inhibitory function of the RIS-Ad. Characteristic nucleotide positions are given above the boxes. (**B**) Schematic presentation of the transgene cassettes of the adenoviral shuttle plasmids tested. Formation of rAdV after transfection of the corresponding PacI-linearized pAdEasy plasmids into HEK293 cells is indicated by “+“ and “−“ signs. (**C**) Anti-Rep Western blot analysis after infection of HEK293 cells for 64 h with rAdV-MMTV-sRep (amplification round 5) and rAdV-M2-Tet-sRep (round 6) at the indicated MOIs (genomic particles/cell). Transfection of pDG plasmid (transfection efficiency about 70%) was used as positive control. (**D**) Anti-Rep Western blot analysis after infection of HEK293 cells with rAdV-MMTV-sRep as in (**C**) in the absence and presence of doxycycline. (**E**) Southern blot analysis for rAAV replicative DNA intermediates after transfection of HEK293 cells with pTR-UF5 and over-infection with rAdV-M2-Tet-sRep in the absence or presence of doxycycline at the indicated MOIs for 64 h. The positive control in lane 2 represents pTR-UF5/pDG co-transfected cells.

**Figure 2 viruses-15-00064-f002:**
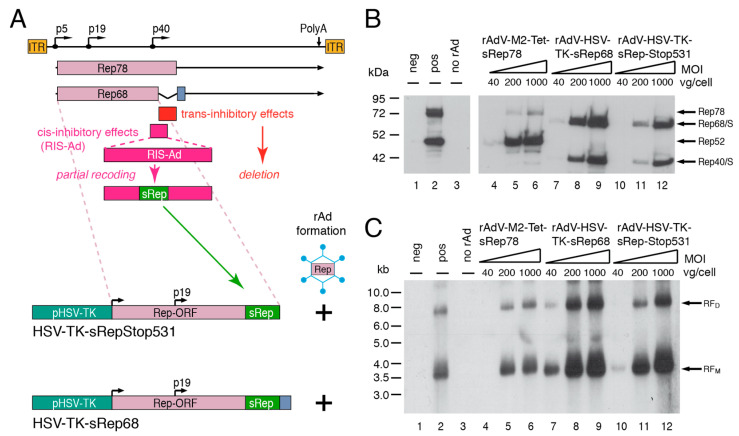
Improved rAdV-mediated expression of functional large Rep proteins by means of the constitutive HSV-TK promoter. (**A**) Genome organization of AAV-2 with a close-up of the RIS-Ad (*Rep inhibitory sequence for adenoviral replication*) in the 3′-part of the rep gene (for further details compare Figure 1A) and depiction of the large Rep protein Rep68 and the truncated Rep-Stop531 protein, which terminates immediately after the major AAV-2 splice donor site. (**B**) Anti-Rep Western blot analysis after co-infection of HEK293 cells with purified rAAV-GFP vector (MOI 100) and either rAdV-M2-Tet-sRep, rAd-HSV-TK-sRep68 or rAd-HSV-TK-sRep-Stop531, as indicated, for 64 h. Positive control are pTR-UF5/pDG co-transfected cells. (**C**) Same experiment as in (**B**) with extraction of viral DNA and Southern blot analysis with a probe for the CMV-GFP transgene cassette. Arrows indicate monomeric and dimeric replicative rAAV-GFP DNA intermediates. Representative data from a total of three experiments is shown.

**Figure 3 viruses-15-00064-f003:**
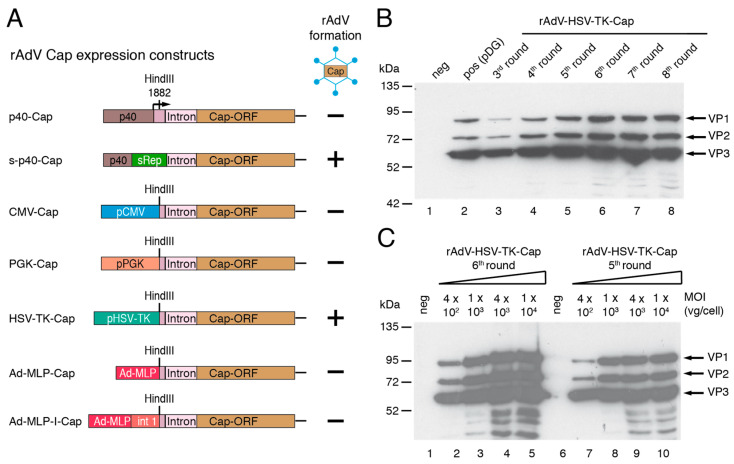
Generation of rAdV for AAV-2 capsid protein expression. (**A**) Schematic presentation of the adenoviral shuttle plasmids. Formation of rAdV after transfection of the corresponding PacI-linearized pAdEasy plasmids into HEK293 cells is indicated by “+“ and “−“ signs. (**B**) Infection of HEK293 cells for 64 h with equal amounts of freeze–thaw supernatants from amplification rounds 3 to 8 of rAdV-HSV-TK-Cap amplification with subsequent anti-Cap western analysis of whole cell extracts. Arrows indicate positions of capsid proteins VP1 to VP3. pDG transfected HEK293 cells were used as positive control (**C**). As in (**B**) with rAdV-HSV-TK-Cap freeze–thaw supernatants from amplification rounds 6 (left lanes) and 5 (right lanes) at different MOIs for 64 h.

**Figure 4 viruses-15-00064-f004:**
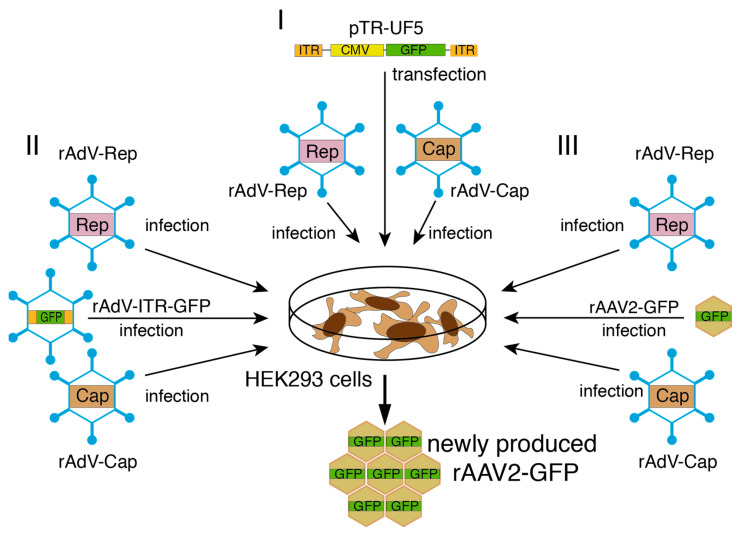
Setups for rAdV-mediated rAAV2-GFP vector packaging. The different experimental approaches for rAdV-mediated packaging of rAAV2-GFP vectors in HEK293 cells are illustrated: (**I**) infection of pTR-UF5 transfected cells with a combination of rAdV-M2-Tet-sRep or rAd-HSV-TK-sRep-Stop531 (generally denoted as *rAdV-Rep* for simplification) and rAdV-HSV-TK-Cap (denoted as *rAdV-Cap*), (**II**) triple rAdV infection with rAdV-Rep, rAdV-Cap and rAdV-ITR-GFP providing the AAV-2 ITR flanked CMV-GFP transgene cassette and (**III**) amplification of low input amounts of rAAV2-GFP by coinfection with rAdV-Rep and rAdV-Cap.

**Figure 5 viruses-15-00064-f005:**
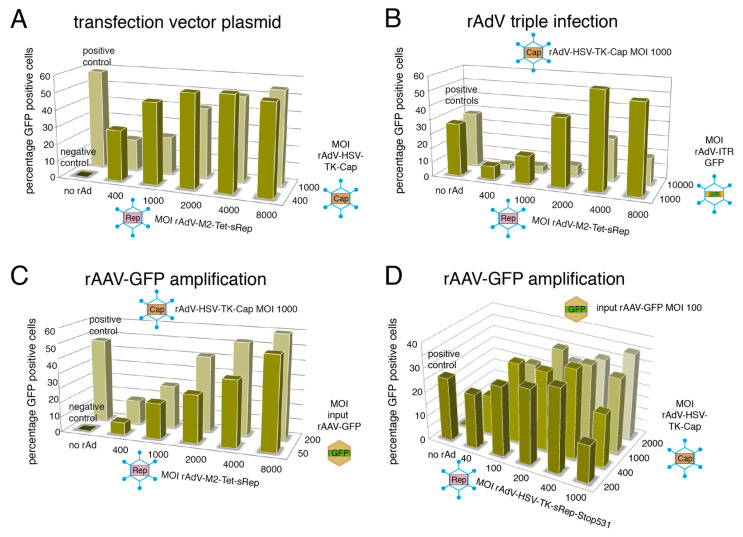
Optimization of rAdV-mediated rAAV2-GFP vector packaging. (**A**–**D**) Percentage of GFP positive cells obtained after transduction of 2 × 10^5^ HeLa cells with small aliquots of freeze–thaw lysates from packaging experiments performed with 5 × 10^5^ HEK293 cells in 6-well plates. For (**A**–**C**), 1/15 shares of the primary freeze–thaw lysates were used for transduction, while in (**D**) 1/30 shares were used. (**A**–**C**) correspond to the setups I to III illustrated in Figure 4 with rAdV-M2-Tet-sRep for expression of Rep proteins, while (**D**) corresponds to the setup III (rAAV-GFP amplification) with rAd-HSV-TK-sRep-Stop531 used for Rep expression. Input MOIs of the different vectors are indicated.

**Figure 6 viruses-15-00064-f006:**
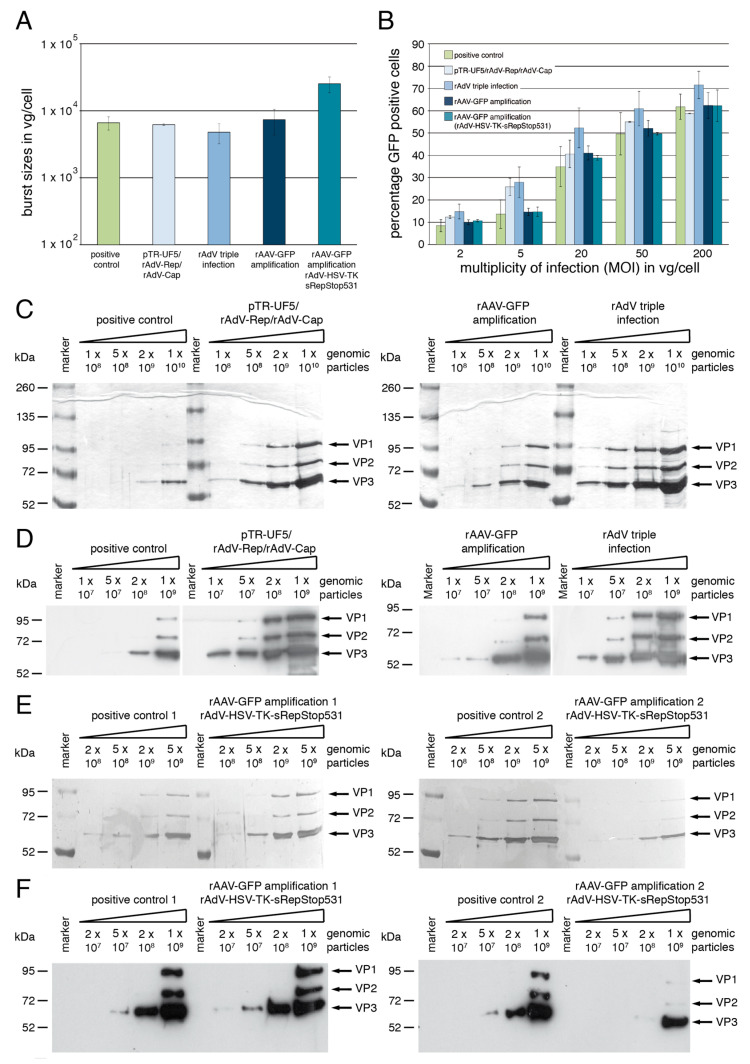
Characterization of affinity-purified rAAV2-GFP vector preparations generated with the help of recombinant adenoviruses. (**A**) Mean burst sizes in genomic particles per cell (vg/cell) for affinity-purified rAAV2-GFP vector preparations generated with different packaging approaches as indicated. The middle three bars involve rAdV-M2-Tet-sRep for expression of Rep proteins, while the right bar shows the preparations from the rAAV-GFP amplification protocol with rAd-HSV-TK-sRep-Stop531. Means were obtained from three preparations for the positive control (pTR-UF5/pDG co-transfection), the rAdV triple infection and the rAAV-GFP amplification mediated by the combination of rAd-HSV-TK-sRep-Stop531 and rAdV-HSV-TK-Cap. For the remaining setups, two preparations were averaged. Standard deviations are indicated by the corresponding error bars. (**B**) Percentage of GFP positive cells after transduction of HeLa cells with 5 different MOIs (expressed in vg/cell) of affinity purified rAAV2-GFP preparations from different setups. Means and standard deviations shown by errors bars were obtained from experiments with three positive control rAAV preparations and two preparations each for the remaining setups described in detail in (**A**) and in the text. (**C**,**E**) Silver staining of SDS PAGE gels loaded with different amounts of genomic particles, as indicated, from selected purified rAAV2-GFP vector preparations. For (**C**) the respective preparations with the highest burst size for the four different packaging methods presented in the first four bars of part A were analyzed, while for (**E**) the two preparations with the highest burst sizes for the positive control and the rAd-HSV-TK-sRep-Stop531/rAdV-HSV-TK-Cap-mediated rAAV amplification were analyzed. (**D**,**F**) Western blot analysis of capsid protein content with different amounts of genomic particles from the same rAAV2-GFP vector preparations as in (**C**,**E**).

**Figure 7 viruses-15-00064-f007:**
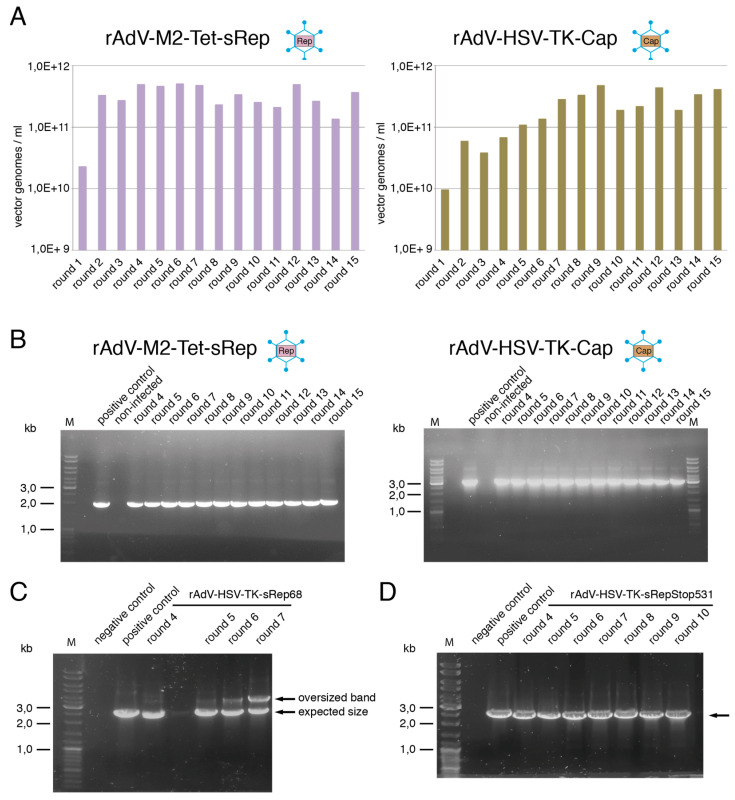
Genomic stability of rAdV used in rAAV packaging. (**A**,**B**) rAdV-M2-Tet-sRep and rAdV-HSV-TK-Cap were amplified in HEK293 cells over 15 round as described in the methods section. (**A**) Determination of genomic rAdV titers. (**B**) Analysis of the genomic integrity by PCR-amplification with oligonucleotides binding at the very ends of the respective transgene cassette. The pAdEasy plasmids used for generation of the corresponding rAdVs were used as positive controls. (**C**,**D**) Analysis of the transgene cassette in supernatants from different amplification rounds of (**C**) rAdV-HSV-TK-sRep68 and (**D**) rAdV-HSV-TK-sRep-Stop531. The oligonucleotides used for PCR amplification bind within the adenoviral sequences flanking the transgene cassette in the pAdEasy plasmids, which were also used as positive controls.

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
