# Peer review of "High-Level rAAV Vector Production by rAdV-Mediated Amplification of Small Amounts of Input Vector"

_viruses, 2022, doi:10.3390/v15010064_

Round 1

Reviewer 1 Report

For a long time, it has been difficult to produce or stably propagate a hybrid recombinant adenovirus carrying AAV rep and cap, which is thought to be useful for rAAV production. This manuscript presented a solution using a pair of rAd vectors to express AAV rep68 and AAV capseparately, by which the cis and trans elements of in the AAV genome that inhibit the production of the hybrid Ad-AAV vectors can be eliminated. Through this approach, high-level rAAV production can be achieved by the amplification of the input rAAV vector in the presence of the co-infection of two rAd vectors expressing AAV rep68 and AAV cap, respectively. Reviewer recommends its publication in MDPI Viruses

There is no major criticism, but one concern that the author should address: the transduction assay of the rAAV vectors (Fig. 6B). As described in the method section, reviewer believes that such AAV vector potency assay was conducted in the superinfection of adenovirus type 2, which initiated the replication of input rAAV vector and thus maximized the transgene expression. This is not the practical method to determine the rAAV vector potency for the application in biomedicine research or gene therapy. If the author can provide the transduction assay without the presence of Ad would be great.

Minors: 

1)    Page 14, lines 479 and 482, Fig. 6B should be Fi. 7B

2)    Page 13, lines 440-442. Can Author detail the way how to detect the wt AAV genomes by PCR-based assay in the method section?

3)    Consistence: page 17, lines 616 and more. Please correct the vg/c to vg/cell as in most the text, vg/cell was used.

Author Response

For a long time, it has been difficult to produce or stably propagate a hybrid recombinant adenovirus carrying AAV rep and cap, which is thought to be useful for rAAV production. This manuscript presented a solution using a pair of rAd vectors to express AAV rep68 and AAV cap separately, by which the cis and trans elements of in the AAV genome that inhibit the production of the hybrid Ad-AAV vectors can be eliminated. Through this approach, high-level rAAV production can be achieved by the amplification of the input rAAV vector in the presence of the co-infection of two rAd vectors expressing AAV rep68 and AAV cap, respectively. Reviewer recommends its publication in MDPI Viruses

There is no major criticism, but one concern that the author should address: the transduction assay of the rAAV vectors (Fig. 6B). As described in the method section, reviewer believes that such AAV vector potency assay was conducted in the superinfection of adenovirus type 2, which initiated the replication of input rAAV vector and thus maximized the transgene expression. This is not the practical method to determine the rAAV vector potency for the application in biomedicine research or gene therapy. If the author can provide the transduction assay without the presence of Ad would be great.

Response: The monomeric rAAV-GFP vectors used in the present work have slow expression kinetics and therefore transduction assays were performed in the presence of adenovirus type 2 to augment second-strand synthesis and obtain readouts for transductions efficiencies in a reasonable time span. Such transduction assays in the presence of adenovirus are well suited to compare different production procedures. With regard to a possible application of this novel rAAV packaging procedure for clinical research, which will be addressed in future studies, I fully agree with the reviewer that the transduction assays should then be performed in the absence of adenovirus and preferentially also in the cell types, which are to be targeted by the vector in vivo in the respective application. However, such clinical applications are out of the scope of the current manuscript. 

Minors: 

1)    Page 14, lines 479 and 482, Fig. 6B should be Fi. 7B

Response: Thank you very much for bringing this to attention. The text was corrected accordingly (lines 499 and 502 of the revised manuscript).

2)    Page 13, lines 440-442. Can Author detail the way how to detect the wt AAV genomes by PCR-based assay in the method section?

Response: A description of the Light-Cycler based Real Time PCR assay was included in the materials and methods section, subsection 2.10, lines 204 to 209.

3)    Consistence: page 17, lines 616 and more. Please correct the vg/c to vg/cell as in most the text, vg/cell was used.

Response: Thank you very much for this notice. The text was corrected accordingly (lines 636 to 641).

Reviewer 2 Report

The study by Weger builds on a body of literature that describes the characterisation of an element at the 3’ of the AAV rep gene that inhibits the replication of adenovirus. The sequence has been termed the Rep inhibition sequence (Ad-RIS) and functions through a yet to be determined mechanism that does not appear to involve poly-peptide or micro-RNA. Armed with this information, the goal of the current study is to build and characterise a series of adenovirus constructs with the intent of generating a platform to package recombinant AAV. This vector system is gaining traction as a powerful gene delivery system with clinical utility. However, one limitation of the vector system is that current packaging platforms are not producing vector at sufficient scale to meet clinical demands to advance translation of the technology. The author hypothesises that the availability of adenovirus-mediated delivery of the different genetic elements required for recombinant AAV production could potentially ease this demand, especially if the platform improves the efficiency of vector production.

Overall, the experimental design is logical and competently executed and the manuscript is generally well-written. The Southern blots, Western blots and silver stains are excellent in quality.  Given that the goal of this study is to build a packaging platform for clinical grade vector production, I draw attention to addressing the following issues ahead of publication to better understand some of the potential limitations of the platform.

1)      Wild-type virus contamination-Lines 440-441 states “AAV-2 wildtype genomes were below the detection limit of the PCR-based assay in all preparations”. However, there is no mention of the detection limit or the methodology used. The author needs to include the methodology and limit of sensitivity of the assay.

2)      Empty-full capsid ratio-The data in Figure 6 allude to the possibility that the preferred combination of rAAV-GFP amplification along with adenovirus delivery of rep and cap sequences generates a larger proportion of empty to full vector capsids than other combinations of experimental conditions. The author should quantitate the relative differences of empty to full capsids generated in the different experimental conditions.

3)      Contamination of recombinant AAV vector stock with adenovirus proteins -a significant advancement in AAV vectorology came in the late 1990s with the cloning of adenovirus helper functions. Use of such packaging plasmids removed the necessity for adenovirus co-infection to generate recombinant AAV. This subsequently led to improved longevity of transgene expression in vivo because pro-inflammatory adenovirus proteins that contaminated some vector preparations were no longer present. The current packaging platform re-introduces adenovirus contamination as a potential phenomenon in vector stocks . While I expect in vivo studies using AAV vector produced by the platform will follow in subsequent manuscripts, the author should address in the discussion section both the potential likelihood for adenovirus contamination using the new platform as well as the consequences should this occur.

4)      Experiment replication-there is no indication in the figure legend as to how many times each experiment was performed. This requires correction.

5)      Figure 6 shows error bars but there is no description in the legend as to how they were generated. This requires correction.

Author Response

The study by Weger builds on a body of literature that describes the characterisation of an element at the 3’ of the AAV rep gene that inhibits the replication of adenovirus. The sequence has been termed the Rep inhibition sequence (Ad-RIS) and functions through a yet to be determined mechanism that does not appear to involve poly-peptide or micro-RNA. Armed with this information, the goal of the current study is to build and characterise a series of adenovirus constructs with the intent of generating a platform to package recombinant AAV. This vector system is gaining traction as a powerful gene delivery system with clinical utility. However, one limitation of the vector system is that current packaging platforms are not producing vector at sufficient scale to meet clinical demands to advance translation of the technology. The author hypothesises that the availability of adenovirus-mediated delivery of the different genetic elements required for recombinant AAV production could potentially ease this demand, especially if the platform improves the efficiency of vector production.

Overall, the experimental design is logical and competently executed and the manuscript is generally well-written. The Southern blots, Western blots and silver stains are excellent in quality.  Given that the goal of this study is to build a packaging platform for clinical grade vector production, I draw attention to addressing the following issues ahead of publication to better understand some of the potential limitations of the platform.

1)  Wild-type virus contamination-Lines 440-441 states “AAV-2 wildtype genomes were below the detection limit of the PCR-based assay in all preparations”. However, there is no mention of the detection limit or the methodology used. The author needs to include the methodology and limit of sensitivity of the assay.

Response: A description of the Light-Cycler based Real Time PCR assay and its limit of sensitivity was included in the materials and methods section, subsection 2.10, lines 204 to 209.

2)  Empty-full capsid ratio-The data in Figure 6 allude to the possibility that the preferred combination of rAAV-GFP amplification along with adenovirus delivery of rep and cap sequences generates a larger proportion of empty to full vector capsids than other combinations of experimental conditions. The author should quantitate the relative differences of empty to full capsids generated in the different experimental conditions.

     Response: The issue of quantitating the relative differences in the contents of empty to full vector particles was addressed in the results of the revised version (lines numbers 466 to 478).

3)  Contamination of recombinant AAV vector stock with adenovirus proteins -a significant advancement in AAV vectorology came in the late 1990s with the cloning of adenovirus helper functions. Use of such packaging plasmids removed the necessity for adenovirus co-infection to generate recombinant AAV. This subsequently led to improved longevity of transgene expression in vivo because pro-inflammatory adenovirus proteins that contaminated some vector preparations were no longer present. The current packaging platform re-introduces adenovirus contamination as a potential phenomenon in vector stocks . While I expect in vivo studies using AAV vector produced by the platform will follow in subsequent manuscripts, the author should address in the discussion section both the potential likelihood for adenovirus contamination using the new platform as well as the consequences should this occur.

     Response: I fully agree with the reviewer that possible contaminations of the final rAAV vector product with Ad or adenoviral proteins are an issue. Possible consequences and measurements are now discussed in lines 662 to 672 of the discussion section.

4)  Experiment replication-there is no indication in the figure legend as to how many times each experiment was performed. This requires correction.

     Response: The Number of experiments (3) is now indicated in the figure legend (line 325).

5)  Figure 6 shows error bars but there is no description in the legend as to how they were generated. This requires correction.

     Response: A description for the generation of the error bars was included in the legend to fig. 6B (lines 427 to 430).